# Physical Activity and Sedentary Behaviour Patterns in 326 Persons with COPD before Starting a Pulmonary Rehabilitation: A Cluster Analysis

**DOI:** 10.3390/jcm8091346

**Published:** 2019-08-29

**Authors:** Wolfgang Geidl, Johannes Carl, Samuel Cassar, Nicola Lehbert, Eriselda Mino, Michael Wittmann, Rupert Wagner, Konrad Schultz, Klaus Pfeifer

**Affiliations:** 1Department of Sport Science and Sport, Division Exercise and Health, Friedrich-Alexander University Erlangen-Nürnberg (FAU), 91058 Erlangen, Germany; 2Clinic Bad Reichenhall, Centre for Rehabilitation, Pulmonology and Orthopedics, 83435 Bad Reichenhall, Germany

**Keywords:** chronic obstructive pulmonary disease, exercise, motor activity, lung diseases, classificatory approach

## Abstract

This study applies a cluster analysis to identify typical physical activity (PA) and sedentary behaviour (SB) patterns in people with chronic obstructive pulmonary disease (COPD) before starting pulmonary rehabilitation (PR). We implemented an observational design which assessed baseline data of objectively measured PA and SB from the STAR (Stay Active after Rehabilitation) study. A total of 355 persons wore an accelerometer (Actigraph wGT3X) for seven days before the start of their PR. Sociodemographic and disease-related parameters were assessed at the start of PR. We applied cluster analysis and compared clusters applying univariate variance analyses. Data was available for 326 persons (31.6% women; age ø = 58 years). Cluster analysis revealed four movement clusters with distinct PA and SB patterns: Sedentary non-movers (28.5%), sedentary occasional movers (41.7%), sedentary movers (19.6%), and sedentary exercisers (10.1%). The four clusters displayed varying levels of moderate PA before rehabilitation (Ø daily min: 9; 28; 38; 70). Notably, all four clusters displayed considerably long average sedentary time per day (Ø daily minutes: 644; 561; 490; 446). The clusters differed significantly in disease-related parameters of GOLD severity, FEV1, CAT, and 6-Min-Walk-Test. In addition to PA promotion, PR programs should consider the reduction of sedentary behaviour as a valuable goal.

## 1. Introduction

Regular physical activity (PA) is proven to help treat noncommunicable diseases (NCDs) [1]. Accordingly, the World Health Organization (WHO) [2] recommends adults with NCD perform at least 150 min of moderate intense aerobic PA per week or 75 min of vigorous intensity aerobic PA for improved health. For people with chronic obstructive pulmonary disease (COPD) engaging in the recommended moderate-to-vigorous physical activities (MVPA) results in improved muscle function, increases in exercise capacity, decreased mood disturbance, reduced symptom burden, and improved cardiovascular function [3].

Nevertheless, health-related movement behaviour is multi-layered. It not only includes MVPA but a wide variety of PA in different intensities spanning from running, walking or standing, as well as sedentary behaviour (SB) [4]. For people with COPD, all aspects of movement behaviour are significant and are associated with several health consequences. PA levels, including PA in light, moderate and vigorous intensities positively influence various health outcomes, including reduced risks of COPD exacerbations, reduced risk of mortality, and improved quality of life [5]. Light intensity PA (e.g., slow walking) seems to be beneficial, which among other benefits leads to reduced risk of COPD hospitalizations [6]. In addition to low levels of PA, SB is now considered a separate risk factor. After adjusting for MVPA levels, sedentary time is a strong predictor of mortality in the general population [7] and in people with COPD [8]. For people with COPD, high sedentary time is associated with poor overall and mental health, unhealthy aging, and overnight hospital stays [9].

Looking at persons with COPD at a group level, average PA levels are low and sitting time is high [10,11,12]. Nevertheless, the heterogeneity in movement behaviours between individuals is remarkably high. Evenson et al. [13] showed large differences in MVPA and sitting time in a sample of 4510 healthy adults. In people with COPD, however, the dominance of means-based analysis in previous research [10,11,12] has yet failed to reveal this heterogeneity in most studies.

Classification approaches (e.g., cluster analysis) are superior to means-based analysis; they help to generate a detailed understanding of movement behaviours by exploring heterogeneity and aiming to find distinct PA and sedentary patterns. In 2017, Mesquita et al. [14] initially applied cluster analysis techniques to movement behaviour in people with COPD. The authors showed that cluster analysis is a powerful classification tool for this clinical population as they identified five distinct movement clusters with significant differences in daily PA within different intensity levels. Their results illustrate that, even if the average PA in people with COPD might be low, specific subgroups may be sufficiently active to meet guidelines. The promotion of physical activity is a central goal of PR [3]; nevertheless, it is unclear whether this goal is equally important for all persons with COPD and how the initial conditions of rehabilitants differ with regard to physical activity levels. In people with COPD before starting a PR, a classification approach which considers all health-relevant intensity ranges of PA and SB is still pending.

Using baseline data from the STAR (Stay Active after Rehabilitation) study, this study quantified health-related PA and SB in patients with COPD, to identify typical movement clusters based on those measures. Secondly, we compared clusters with regard to demographic and disease-related characteristics.

## 2. Methods

This study uses cross-sectional data from the baseline assessment of the STAR study participants (Clinical Trials Registration Number Clinicaltrials.gov, ID: NCT02966561) [15]. The STAR study protocol was approved by the independent Research Ethics Committee of the Medical Faculty of Friedrich-Alexander University Erlangen-Nürnberg, Germany in 2015 (Re.-No. 321_15B).

### 2.1. Study Population

This study includes subjects who had been referred by their doctor to a PR in the inpatient pulmonary rehabilitation clinic Bad Reichenhall (Bavaria, Germany) with the following inclusion criteria: Physician-confirmed diagnosis of COPD (international classification code: J44.- at all 2011 GOLD classifications A–D); age ≥ 18 years. We excluded persons in case of severe comorbidities or inability to speak German. The PR was paid for by the German pension insurance fund for roughly 90% of all participants. Therefore, most of the patients with COPD were employed and younger than 65 years. The rehabilitation team at clinic in Bad Reichenhall contacted a total of 797 persons approximately one month before their planned three-week inpatient stay in the rehabilitation clinic, asked via telephone whether they would be willing to voluntarily join the study. Of these, 418 patients (52.4%) provided informed consent to study participation and subsequently received a plain language statement, a questionnaire and a hip-worn accelerometer (ActiGraph wGT3X-BT, Pensacola, FL, USA). A total of 92 participants were excluded from the analysis due to the following reasons: Retraction of the initial COPD diagnosis of the family doctor by the lung specialist in the rehabilitation clinic (*n* = 62), persons not attending PR (*n* = 1), retrospective withdrawal of consent to data use (*n* = 1), unreturned Actigraph device or insufficient wear time (*n* = 28). This led to an overall sample of 326 patients included in the activity analysis (see Figure 1 and Table 1).

### 2.2. Assessments

#### 2.2.1. Physical Activity and Sedentary Behaviour

Daily PA and SB were objectively assessed using the validated tri-axial Actigraph wGT3X-BT accelerometer [16,17] which is explicitly recommended for subjects with COPD [18,19]. Participants were instructed to wear the activity monitor on the right hip for seven days during waking hours and reminded to take off the device only during water-based activities [17,20]. Wear-time and periods of non-wear time, e.g., taking a shower, were logged at the end of each day in the activity diary provided. Participants were sent the activity monitors two weeks before their rehabilitation was scheduled and asked to wear the device for the seven-day measurement under free-living conditions in their home before entering the rehabilitation clinic.

#### 2.2.2. Secondary Outcomes

All secondary outcome measures were completed upon arrival at the inpatient pulmonary rehabilitation clinic Bad Reichenhall. Secondary outcome measures included the COPD assessment test (CAT) [21]. The CAT is a short, simple questionnaire consisting of eight statements that measure disease impact on health status. Scoring for the test ranges from 0–40 with higher scores denoting higher levels of disease impact. Scores < 10 indicate a low impact, 10–20 medium, 21–30 high, and >30 a very high impact level. Patients’ functional capacity was assessed with one 6-minute walk test (6MWT). The 6MWT is a valid, responsive and reliable test that is considered representative of the overall functional status for people with COPD [22]. Finally, airflow limitation was measured in the clinic using post-bronchodilator FEV1/FVC to classify patients into the four GOLD stages; 1, mild; 2, moderate; 3, severe; and 4, very severe [23]. Furthermore, GOLD classification A/B/C/D was extracted from the information on the individuals’ exacerbations, clinic stays, and CAT scores [24]. Finally, the number of comorbidities was based on a standardised list of 31 standardized diseases and eight free specifications filled out by the responsible physician.

### 2.3. Assessments

#### 2.3.1. Processing of Accelerometer Data

We used the ActiLife v6.13.3 software to transform the raw accelerometer output into cumulated activity scores attributable to different intensity categories. We thereby relied on the Freedson algorithm [25] which defines fixed count values in accordance with established metabolic equivalent tasks (MET) cut-offs: Sedentary behaviour with activity counts of 0 to ≤100 per minute, light PA with activity counts >100 to 1951, moderate PA with activity counts >1952 to 5724 (MPA), and vigorous physical activity (VPA) with activity counts > 5725 [20,26]. In order to ensure differentiation between very light forms of PA (1.5 MET < x < 2 MET) and light forms of PA (2 MET ≤ x < 3 MET), the Freedson cutpoints were supplemented by the additional cutpoint of 929 points proposed by Cain and Geremia [27]. According to the methodological standards proposed by Byrom and Rowe [20]. Actigraphs were initialised at a frequency of 100 Hz and downloaded using 15 second epochs. A measurement was considered valid if the patients had a wear-time of ≥10 h per day for at least five of the seven measuring days with no requirements for specific numbers of weekend or week days [20]. For valid measurements, all available measuring days with a wear-time of ≥10 h were taken into account in the analysis.

#### 2.3.2. Determining the Number of Clusters

To determine the number of clusters, we firstly explored the time participants spent in the different intensity categories. Across the entire sample, participants scarcely recorded activity in the vigorous intensity range; the average daily activity time in vigorous PA was less than one minute. Against the background of the volume of VPA near to zero, we heavily questioned the suitability of VPA as an indicator for the identification of clusters. Our assumption was confirmed by principal component analysis (PCA) which revealed comparably low anti-image correlations for the VPA parameter. Hence, we have not included the parameter VPA for the cluster formation. We relied on the cluster formation process of four z-standardized activity indicators: Average daily sedentary time, and PA in very light, light and moderate intensity (see Appendix A).

Hierarchical cluster analysis was applied to determine the final number of clusters. We relied on squared Euclidean distance metric and the ward algorithm as a conservative merging procedure [28]. In the scientific literature, there is no formal gold standard of how to assign the number of clusters, instead a combination of statistical and content-related arguments should be considered [28]. Accordingly, we first inspected the Scree plot (see Appendix A) which visualises the increase in explained variance while reducing the number of hypothetical clusters. In addition to this subjective decision procedure, we made use of two stopping rules which have achieved the best results in an extensive simulation study [29] comparing a total of 30 indicators: The Calinski and Harabasz [30] criterion and the Je(2)/Je(1) criterion by Duda and Hart [31].

#### 2.3.3. Comparison of the Different Clusters

After the cluster formation process, we extracted descriptive statistics for the characterization of different activity clusters including time spent in SB, very light PA, light PA, moderate PA, vigorous PA and, finally, the total number of steps per day.

Afterwards, the different clusters were compared with respect to the following information from the baseline assessment: Gender (male vs. female), age (in years), respiratory function (via FEV1 which forms the basis for the classification of disease severity), body composition (body mass index (BMI; in kg/m^2^), extracted objectively from measured height and weight), exercise capacity (six-minute-walking-test (meters)), and disease impact (COPD Assessment Test, CAT, range: 1–40). In addition, we performed an activity bout analysis which quantifies time spent continuously in specific intensity areas (similar as to McVeigh et al. [32]).

All analyses were run in the software R, version 3.4.3 using the package NbClust [33]. For the cluster comparison, we calculated univariate ANOVA with the effect size η^2^. If necessary, TukeyHSD post-hoc analyses served to attribute an overall effect to concrete cluster constellation(s). When the assumption of variance homogeneity was violated, we applied Welch’s ANOVA and the Dunnett post-hoc test. For the dichotomous variable of gender, we used the chi-square (χ^2^) test to inspect substantial deviations from equal distribution across the clusters. The significance level was set at *p* < 0.05.

## 3. Results

### 3.1. Number of Different Movement Behaviour Clusters

The scree plot (see Appendix A) suggested that the activity indicators can be adequately described by three clusters. The Calinski–Harabasz criterion favours three clusters (CHmax = 279.44) whereas the Duda–Hart criterion suggest an extraction of four clusters (DHmax = 0.688). To clarify these ambiguous recommendations, we undertook a deeper comparison of both solutions to reach a final decision on the number of clusters. The analysis revealed that the three and four cluster options were nested within each other, with the slight distinction that the four cluster solution still differentiates within the most active cluster at the top of the activity scale. Due to this finer extraction, we present results from the four cluster solution.

Overall, the clusters explained between 50% and 76% of the variance in the four indicators, i.e., sedentary behaviour, F(3, 112.8) = 94.5, *p* < 0.001, η^2^ = 0.501, very light PA, F(3, 106.6) = 123.8, *p* < 0.001, η^2^ = 0.533, light PA, F(3, 106.7) = 265.0, *p* < 0.001, η^2^ = 0.760 and moderate PA, F(3, 102.9) = 193.3, p < 0.001, η^2^ = 0.600. Importantly, there was no significant difference in the accelerometer wear time between the different clusters, F(3, 322) = 2.46, *p* = 0.063.

### 3.2. Characterisation of the Clusters: Physical Activity and Sedentary Behaviour

The cluster formation process resulted in four distinct clusters of different size. Cluster one (sedentary non-movers) comprised 93 individuals (28.5%) who were comparably inactive (e.g., an average of 644.3 ± 76.0 min sedentary behaviour per day) and in most cases (94.6%) did not meet the WHO guidelines for PA [2]. With a total of 136 persons, cluster two (sedentary occasional movers) was the largest cluster (41.7%) and somewhat more active than the first cluster (e.g., an average of 561.5 ± 59.9 min sedentary behaviour per day). Cluster three (sedentary movers) included 64 individuals (19.6%) who were comparably active in relation to 70.2% of patients made up of clusters one and two. They performed an average of 38.9 ± 12.0 min MVPA per day. Correspondingly, the large majority of this cluster (95.3%) fulfilled the aforementioned activity guidelines. Finally, cluster four constitutes the smallest (10.1%) but by far most active cluster. The 33 members had a daily sedentary time of 445.6 ± 68.6 min, but more importantly, they performed an average of more than one hour MVPA per day (71.4 ± 25.4 min). According to the cluster designations of Mesquita et al. [14] we have named cluster 3 sedentary movers, and cluster 4 sedentary exercisers. Overall, there was a progressive increase for activity indicators across the four clusters (see Table 2 and Figure 2).

### 3.3. Characterisation of the Clusters: Other Parameters

There was a main effect for the age variable, F(3, 322) = 2.70, *p* = 0.046, η^2^ = 0.025. Post-hoc analyses revealed that cluster one was significantly older than cluster three (d = 0.43). Gender was equally distributed across the four clusters, χ^2^(3) = 2.70, *p* = 0.74. Likewise, there were no differences for BMI between the clusters, F(3, 312) = 0.012, *p* = 0.998.

Furthermore, the ANOVA demonstrated that activity clustering is also closely related to functional parameters (see Table 2). For instance, lung function differed significantly across the sample, F(3, 113.0) = 25.61, *p* < 0.001, η^2^ = 0.175. All clusters differed significantly from each other (except the clusters three and four), whereby the FEV1 value was lowest in cluster one and highest in cluster four. In accordance with the fact that the FEV1 parameter forms the basis for the classification of disease severity, there was a similar pattern for GOLD stage classifications 1–4, χ^2^(9) = 49.08, *p* < 0.001, Cramérs V = 0.226. In contrast, no differences could be registered for GOLD classification A/B/C/D, χ^2^(6) = 5.81, *p* = 0.45, and the number of comorbidities, F(3, 322) = 1.05, *p* = 0.371. However, the four clusters also differed significantly with regard to the CAT Score, F(3, 118.7) = 12.78, *p* = 0.998, η^2^ = 0.087. Specifically, the least active cluster perceived a higher impact of disease than the three other clusters.

### 3.4. Activity Bout Analysis

In accordance with the analysis of overall activity time, most bouts were identified as SB. Figure 3 displays the activity bout analysis for all four clusters. The distribution of different bout length categories was broadly comparable across the four clusters. However, the most active clusters (clusters 3 and 4) showed a trivial number of very long (≥60 min) sedentary periods (1.8% and 1.6%) whereas the most inactive clusters spent more than 150 min per day in these segments. The differences in very light activity bouts were even more apparent. In cluster one, time spent in very light PA bouts was negligible compared to their sedentary bouts. Clusters three and four, in contrast, spent a substantial amount of time in these light PA blocks of under five minutes (13.2% and 15.3%) or five to ten min (10.2% and 12.7%). In cluster four, notably, the absolute time in 5–10 min bouts was even higher for very light PA than for SB (11.2%).

Clusters three and four showed only few continuous activity periods of moderate or vigorous intensity. Nevertheless, members of clusters three (M = 41 min; 1.9%) and four (M = 75 min; 3.4%) demonstrated substantially more short bouts (<5 min) of moderate intensity than their counterparts in the cluster one (M = 10 min; 0.3%).

## 4. Discussion

This study provides the first detailed description of health-related movement behaviours in patients with COPD before starting their PR. It is novel in its analysis of movement behaviours across the whole spectrum, from sedentary behaviour to very light, light and moderate intensity PA, including the pattern of these movements in terms of bout duration. Using cluster analysis, our classification approach identified subgroups of patients with COPD with notable different PA and sedentary patterns.

### 4.1. Cluster Differences Regarding PA

Patients with COPD in our study showed markedly different levels of PA before rehabilitation. MVPA is regarded as decisive for health in previous PA recommendations; for example, the WHO guidelines [2] recommend at least 150 min of MPA per week. Our three most active clusters (cluster 2, 3 and 4) had average daily MPA volumes of 28, 38 and 70 min, respectively. Thus, about 70% of all participants with COPD are, according to the guidelines, sufficiently active before rehabilitation starts. The daily step counts in our study ranged from 2749 (sedentary non-movers), to 5649 (sedentary occasional movers), to 7866 (sedentary movers), to 11,045 (sedentary exercises). Depew et al. [34] reported COPD patients need to achieve >4580 steps per day to avoid severe physical inactivity (SPI), and later determined that 69% of patients do not reach this cut-off [35]. In our study, only sedentary non-movers, which includes roughly 30%, remained below this cut-off value. In comparison to other studies in the rehabilitation context (e.g., Spruit et al. [10]), our sample, and especially the three active clusters (2,3,4), were considered sufficiently active before PR. When considering the findings of various reviews reporting low PA levels in people with COPD [10,11,12], this is a rather surprising result. One reason for this finding could be the fact that a large proportion of rehabilitants in our study are still of working age. Rehabilitation was paid for by the German pension insurance for roughly 90% of all participants. Therefore, most of our COPD patients (75%) were still gainfully employed with an average age of just 58 years. Our sample differs from many other COPD samples, most of which are much older and, to a large extent, already retired.

In line with the cluster analysis conducted by Mesquita et al. [14], our study also showed a high heterogeneity and clear differences in the movement behaviour of people with COPD. Means-based analysis often suggest that people with COPD are highly physically inactive. Our results confirm the results of Mesquita et al. [14] that patients with COPD are not physically inactive per se; on the contrary, there are subgroups that are remarkably active. However, Mesquita et al. [14] described different movement clusters, including persons with high volume of MVPA but low volume of light intensity PA; but there are also clusters that behave exactly in opposite directions, i.e., perform a lot of light activities and little MVPA. This is in contrast to our results. For our four clusters, the duration of PA in the three different intensity ranges developed in the same direction, indicating that an increase in one intensity category was also consistently associated with an increase in duration in the other two intensity categories.

Several current PA guidelines, e.g., the recommendations of the American College of Sports Medicine [36], those of the WHO [2] or the German PA guidelines [37], recommend that overall MVPA should be accumulated in bouts of 10 min or longer (e.g., at least 3 × 10 min/day on five days of the week). By contrast, the current US guidelines [38], for example, do not specify the required minimum duration of individual bouts. For nine out of 10 patients with COPD in our study, PA bout lengths of at least 10 min do not occur at all. Van Remoortel et al. [39] previously demonstrated that when PA data is analysed using these bout cut-points, reported MVPA among individuals with COPD is three to 12-fold lower [39]. This observation also applies to our study where longer bouts of MVPA (of at least 10 min as per the WHO guidelines) were rarely observed. Low physical capacity, dyspnea and fatigue make it more difficult to be physically active for longer durations. However, even patients with higher physical capacity (clusters 3 and 4) displayed few bouts of longer duration > 10 min. Attempting to perform PA ’as fast as possible’ to alleviate the discomfort caused by PA [11], a lack of motivation and a lifestyle in which a high proportion of activities of daily living are spread across the day could also be reasons for shorter bouts of MVPA.

### 4.2. Cluster Differences Regarding Sedentary Behaviour

To our knowledge, this is the first study in the field of COPD applying an analytical approach which ensures differentiation between SB (≤ 1.5 MET) and very light forms of PA (1.5 MET < x < 2 MET). This allows a precise analysis of total SB and a distinction from standing. Importantly, all four clusters display a considerable long sedentary time per day (7.5–10.75 h). For this reason, all four cluster designations begin with sedentary. Regardless of PA level or GOLD stage, people with COPD spend a large part of the day in SB before PR. This sedentary time is comparable to SB in other clinical populations, e.g., people with stroke (10.9 h/day) [40], type-2 diabetes (8.0 h/day) [41], multiple sclerosis (8.0 h/day) [42] or coronary artery disease (8,0 h/day) [43].

Long periods of sitting, the most prevalent of SB, is particularly harmful to health if the persons are not also considerably physically active. Ekelund et al. [7] demonstrated in their meta-analysis, including data from more than one million individuals from the general population, that long sitting time is associated with increased risk of mortality. The effect is more pronounced in the more inactive and only high levels of PA (60–75 min of MVPA) seem to eliminate the increased mortality risk. In patients with COPD, objectively measured SB has a prognostic value; Furlanetto et al. [8] previously calculated that individuals with COPD who spent >8.5 h/day in SB had a fourfold increased mortality risk compared to the less sedentary group. In our study, 70% of all participants show an average daily sedentary time of more than 8.5 h and even the most active cluster 4 exhibited an average daily sedentary time of nearly 7.5 h (445 min). Importantly, the sedentary non-movers have an average sedentary time of 10.75 h (644 min).

In addition to the average duration, it is worth considering the manner in which sedentary time is accumulated. Uninterrupted sedentary bouts seem to be particularly deleterious for health [44]. Breaking up prolonged sitting, with moderate intense PA (walking) or even with very light intense PA (e.g., standing) is beneficial to health and leads, among other things, to reduced postprandial glucose and insulin responses [45]. Our bout analyses showed that there are significant differences in the accumulation of sedentary time between the clusters. While the sedentary non-movers and the sedentary occasional movers accumulate their high sedentary time over fewer longer individual bouts, the more active sedentary movers and the sedentary exercisers interrupt their long daily sedentary time more frequently. Therefore, it can be assumed that the more active clusters are more successful in temporally interrupting their very long sedentary periods, which may further enhance the health-related gap between the clusters.

### 4.3. Cluster Differences Regarding Clinical and Sociodemographic Parameters

The four clusters showed no statistically significant differences regarding body weight, distribution of men and women. Gold severity classification (1–4) differed in the clusters with higher disease severities in the less active clusters but the A–D Gold classification did not differ between clusters. The number of comorbidities also does not differ significantly between the clusters. However, differences between the four clusters with regard to disease severity, functional capacity, lung function, and impact of the disease are significant. The following relationship applies to all clusters. On average, if one cluster is compared with the slightly less active cluster (e.g., cluster 1 with cluster 2), the more active cluster shows better functional capacity (six-minute walking test), better lung function (FEV1) and better quality of life (CAT). The differences between the most active and least active cluster are large; six-minute walking test: 503 m vs. 377 m; FEV1: 65% vs. 43%; CAT: 20.6 vs. 26.7. Looking at the CAT scores, mean scores >20 for all four clusters indicate high level of symptom burden across all four clusters [46,47,48]. Due to the cross-sectional design, however, we cannot make any statements about causal relationships. It remains unclear whether lower PA levels are the cause of disease progression and poor lung function; or vice versa with deteriorations of the disease leading to low PA. The two variables probably influence each other, as the “dyspnea-inactivity vicious circle” modelled by Ramon et al. [49] shows.

### 4.4. Implications for Pulmonary Rehabilitation

This study has some important implications for PR programs. First, it underlines that promoting PA must be considered a central goal of PR [3]. Based on our results, promoting a minimum amount of PA does not seem to be equally important for all participants. For the sedentary non-movers promoting quantity of PA remains indeed a key objective. The other three clusters, corresponding to 70% of the sample, largely meet the current PA guidelines regarding the quantity of PA. Nevertheless, in addition to the quantity of PA, qualitative aspects of PA participation experiences are also important for the health effect of PA [50]. Qualitative aspects of PA include, for example, whether PA also increases psychological well-being. Another qualitative aspect of PA can be whether the activity is suitable to contribute to coping with the disease. Certain activities can help people with a chronic condition to distract themselves from the stress of their illness or help people to perceive themselves as “normal” or able to perform despite their condition. For persons meeting the quantity of PA guidelines, rehabilitation should concentrate on the promotion or enhancement of qualitative aspects of PA. Second, for the sedentary non-movers, which are characterised by low fitness, decreased lung function and low PA levels, the current recommendations targeting at least 150 min of weekly MPA, may be overstretching for many [51]. Moy et al. [52] reported that any amount of time spent in MVPA after a hospital stay was associated with reduced risk of mortality among COPD patients. Thus, each additional increase in PA away from inactivity is significant, and important health effects may be experienced even at low levels of activity [53]. The goal of rehabilitation with regard to the PA promotion should be carefully defined here, preferably together with the patient. A good idea is the phased approach proposed by Blondeel et al. [54], which first achieves an improvement in physical performance with an exercise intervention and then thinks about behavioural changes in the next step.

PR programs often fail to achieve sustained and significant increases in PA [55,56]. Individualised, tailored exercise programs enhance the adherence to PA [36]. For patients with COPD this tailoring must take into account the significant differences in movement behaviour, and associated with this the large differences in fitness and lung function.

Considering the high sedentary time of most patients with COPD, it is vitally important that reducing SB plays a larger role during PR and treatment of the disease. For all clusters, reducing sitting time and interrupting long periods of sitting emerges as an important goal. For inactive adults, the meta-analysis from Ekelund et al. [7] suggests that replacing sitting time with light-intensity PA reduces the risk of all-cause mortality. In addition to reducing the volume of SB, preliminary evidence suggests that breaking up prolonged periods of SB with light PA may have positive health effects amongst physically inactive individuals [57]. These conclusions support the statements by Hill et al. [44] who emphasised that SB can be substantially replaced by light PA, thus leading to an overall increase in PA. Increases in light PA include activities of daily living such as going shopping on foot or working in the garden, and represent a more realistic approach to increasing PA in this population. Indeed, the idea that focusing on reducing SB and, therefore, increasing light PA may be more achievable and feasible than increasing MVPA levels is well-supported [12,58]. Clinical COPD guidelines, however, have so far hardly reflected the topic of SB [59]. In the future development of guidelines and PR programs, the topic of reducing SB time should play a greater role. Thus, the “move more and sit less” public health strategy could equally target adults with and without COPD.

### 4.5. Future Research

Our study used a classification approach on the cross-sectional baseline data set of the STAR study. The STAR study also collects PA and SB six weeks and six months after PR. Further classification approaches to this longitudinal data (e.g., Linking of Clusters after Removal of a Residude-analysis; LICUR) will show how the clusters analysed here change their PA and SB after PR. Such analysis, in combination with the measurement of psychological and physical determinants of PA behaviour, will enhance our understanding of patients with COPD movement behaviour and its changes over time.

Basic research should focus further on SB bouts and breaks, and their impact on health in this population. Future applied research should focus on reducing overall time in sedentary behaviour, breaking up long periods of sedentary behaviour, and in increasing PA in light intensity activities.

### 4.6. Limitations

Despite this novel data providing relevant and important insight into PA and SB behaviours in individuals with COPD, some limitations must be discussed. Firstly, although the use of accelerometers has become the scientific standard and this study followed recent data collection and processing recommendations from Byrom et al. [20], it must be noted that there remain limitations to accelerometry: Water-based PA (e.g., swimming or water aerobics) cannot be measured; a valid daily wear time of at least ten hours does not cover the entire 24 h day; this method under-detects non-ambulatory PA (e.g., strength exercise of the upper limbs on a stationary device). Secondly, influences of daylight [60] and meteorological factors [61] have not yet been taken into account in the analyses. Thirdly, only 418 from 797 individuals were willing to participate in this study. It may be that those who participated were more active and motivated with a greater interest in PA than those who refused to participate, which created a selection bias by overestimating the activity levels of the entire sample. Finally, results of this study came from a single clinic and country which may limit the international generalizability of the present findings [62].

## 5. Conclusions

A key contribution of this study is the detailed reporting of health-related movement behaviours in individuals with COPD including SB and PA in very light, light and moderate intensity areas as well as patterns of activity bouts. The present study concludes that patients with COPD perform varied levels of PA in free-living conditions. However, most patients suffering from COPD spend a considerable and unhealthy proportion of their daily lives in SB. Consequently, PR programs should not only aim at the promotion of PA [3] but also consider the reduction of SB as a valuable goal.

Our study proves that cluster analysis of accelerometer data on PA and SB has the potential to identify subgroups of COPD patients with distinct health-related movement patterns. This more comprehensive approach of analysing PA data provides a better understanding of movement behaviours in patients with COPD. Finally, the findings of this study may enable future researchers and clinicians to better plan and individualise PA promotion and SB reduction interventions for COPD populations.

## Figures and Tables

**Figure 1 jcm-08-01346-f001:**
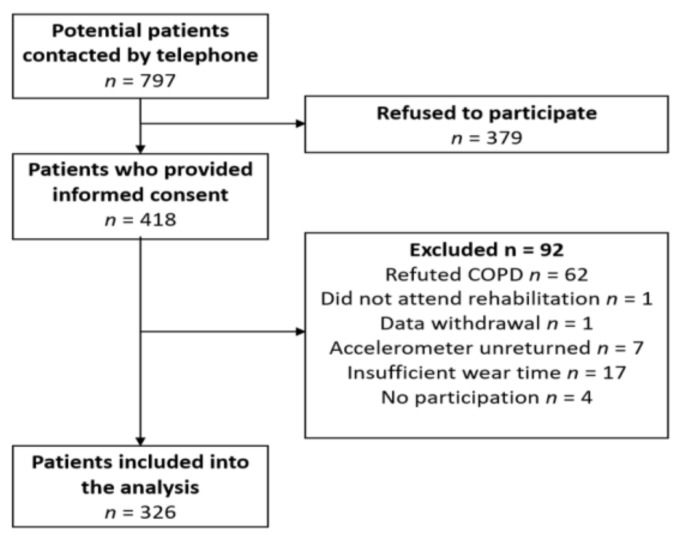
Flow chart for the analysis.

**Figure 2 jcm-08-01346-f002:**
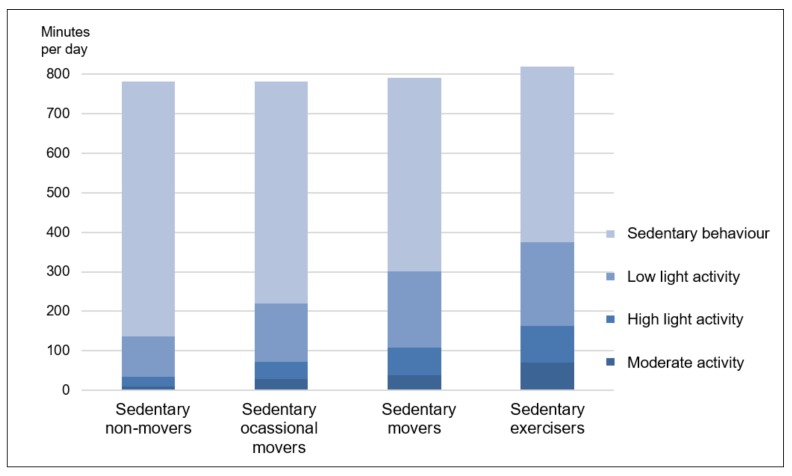
Daily time of the four clusters in sedentary time and in activities of low light, high light and moderate intensity.

**Figure 3 jcm-08-01346-f003:**
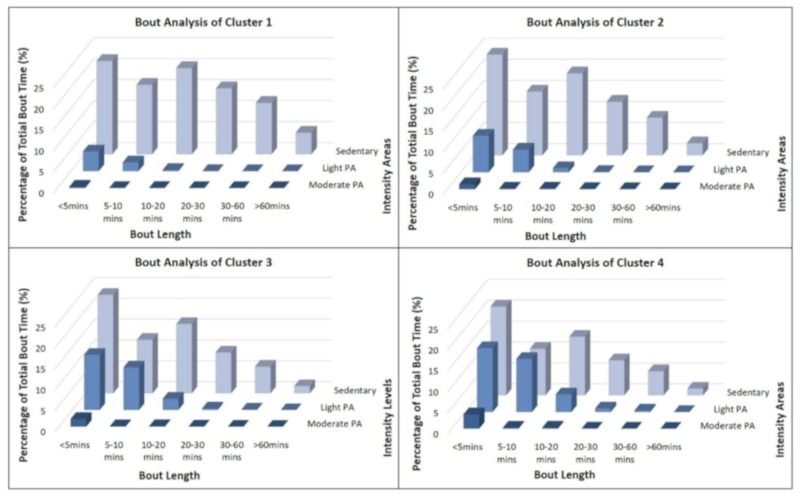
Physical activity and sedentary bout analysis for all four clusters.

**Table 1 jcm-08-01346-t001:** Characteristics of the whole sample.

*n*	326
Age (years)	58.2 ± 5.6
Sex (% male)	67.8%
Height (cm)	170 ± 9.2
Weight (kg)	80.2 ± 20.9
BMI (kg/m^2^)	27.4 ± 6.5
FEV1 (%)	53.7 ± 18.2
GOLD Severity Classification 1/2/3/4 (%)	9.3/44.4/38.0/8.3
GOLD Classification A/B/C/D (%)	1.7/44.2/0/53.6
COPD Assessment Test (CAT Score)	23.37 ± 6.71
Number of Comorbidities	4.48 ± 2.58
Percentage of Current Smokers	45.8%
Employed	75.3%
Sedentary Behaviour (min/day)	559.4 ± 92.9
Moderate-to-Vigorous Physical Activity (min/week)	204.3 ± 160.9
Number of Steps (per day)	5803 ± 3051

**Table 2 jcm-08-01346-t002:** Characteristics of the four clusters.

	Cluster 1 (Sedentary Non-Movers)	Cluster 2 (Sedentary occasional Movers)	Cluster 3 (Sedentary Movers)	Cluster 4 (Sedentary Exercisers)	df	F	*p*	Effect Size η^2^
**General Characteristics**								
N	93	136	64	33				
%	28.5%	41.7%	19.6%	10.1%				
Age (years)	59.5 (6.1)	58.0 (6.0)	57.1 (4.1)	57.4 (4.5)	3, 322	2.70	0.046	0.025
Sex (% male) ^a^	72.0	68.1	64.1	65.6	3	1.24 ^a^	0.744 ^a^	-
BMI (kg/m^2^)	27.39 (7.27)	27.44 (6.68)	27.40 (5.81)	27.19 (5.14)	3, 312	0.012	0.998	-
FEV1 (%)	43.10 (13.81)	54.25 (17.95)	61.82 (17.47)	64.89 (16.71)	3, 113.0	25.61	<0.001	0.175
GOLD 1/2/3/4 (%) ^a^	0/31.8/50.0/18.2	7.7/46.9/39.2/6.2	19.4/50.0/29.0/1.6	21.2/57.6/18.2/3.0	9	49.082 ^a^	<0.001	V = 0.226 ^a^
GOLD A/B/C/D (%) ^a^	0/37.6/0/62.4	1.6/49.2/0/49.2	1.8/49.1/0/49.1	3.3/46.7/0/50.0	6	5.81 ^a^	0.445	-
Comorbidities	4.37 (2.77)	4.29 (2.47)	4.92 (2.60)	4.76 (2.42)	3, 322	1.05	0.371	-
COPD Assessment Test	26.37 (5.84)	22.34 (6.74)	22.63 (7.37)	20.59 (4.68)	3, 118.7	12.76	<0.001	0.087
6-Min-Walk-Test (in m)	386.1 (103.8)	458.7 (97.6)	496.7 (71.5)	503.3 (79.1)	3, 298	22.23	<0.001	0.183
**Physical Activity and Sedentary Behaviour (min/day)**								
Very Light PA	102.97 (26.78)	147.82 (32.29)	191.61 (35.20)	210.15 (55.3)	3, 106.6	123.8	<0.001	0.533
Light PA	25.00 (10.89)	44.81 (10.95)	70.22 (12.46)	93.92 (18.55)	3, 106.7	265.0	<0.001	0.760
Moderate PA	9.23 (6.09)	27.60 (16.12)	38.49 (11.88)	69.76 (23.38)	3, 102.9	193.3	<0.001	0.600
Vigorous PA	0.13 (0.21)	0.34 (1.50)	0.36 (0.88)	1.64 (3.76)	3, 94.4	3.94	0.011	0.067
Steps (per day)	2749 (1064)	5649 (1826)	7866 (1786)	11045 (2621)	3, 106.1	243.3	<0.001	0.678
Overall Sedentary Time	644.33 (76.02)	561.46 (59.93)	490.22 (60.43)	445.58 (68.54)	3, 112.8	94.5	<0.001	0.501
**Wear Time (min/day)**	780.1 (87.2)	781.4 (73.7)	793.1 (82.2)	819.7 (68.0)	3, 322	2.46	0.063	-

Note: Mean values (and standard deviation in brackets) of the different variables; ^a^ Due to the dichotomous character of the variables gender and GOLD classification, we used here the χ^2^-test with the effect size Cramer’s V.r clusters in sedentary time and in activities of very light, light, and moderate intensity.

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
