# Peer review of "Physical Activity and Sedentary Behaviour Patterns in 326 Persons with COPD before Starting a Pulmonary Rehabilitation: A Cluster Analysis"

_jcm, 2019, doi:10.3390/jcm8091346_

Round 1

Reviewer 1 Report

In the present study, the authors analyzed the presence of physical activity and/or sedentary behavior in COPD patients before starting pulmonary rehabilitation. The study was conducted on 326 COPDers, who were grouped according to accelerometer data recorded for 7 days before pulmonary rehabilitation. Patients were grouped as sedentary non-movers, sedentary occasional movers, sedentary movers, sedentary exercisers. The 4 groups displayed different COPD-related parameters. The picture described is interesting. However, what is unclear is the meaning of it, in other words, and so what...

In reality, the reader wants to know whether results and compliance with pulmonary rehabilitation are different in the 4 groups described. In this case, this type of grouping has sense. If not, why should I ask patients to wear for one week an accelerometer? The authors are suggested to provide this information, which they must have.

Author Response

Reviewer 1: 

In the present study, the authors analyzed the presence of physical activity and/or sedentary behavior in COPD patients before starting pulmonary rehabilitation. The study was conducted on 326 COPDers, who were grouped according to accelerometer data recorded for 7 days before pulmonary rehabilitation. Patients were grouped as sedentary non-movers, sedentary occasional movers, sedentary movers, sedentary exercisers. The 4 groups displayed different COPD-related parameters. The picture described is interesting. However, what is unclear is the meaning of it, in other words, and so what...

In reality, the reader wants to know whether results and compliance with pulmonary rehabilitation are different in the 4 groups described. In this case, this type of grouping has sense. If not, why should I ask patients to wear for one week an accelerometer? The authors are suggested to provide this information, which they must have.

—> Dear reviewer, Thank you very much for reading and reviewing our manuscript. We kindly appreciate the evaluation of our article.

We completely agree with your statement. As the longitudinal group-based analysis of changes in physical activity and sedentary behaviour after rehabilitation is the overriding goal of this study, we asked participants in the STAR study to wear the accelerometer again 6 weeks and 6 months after rehabilitation. The current paper represents the first important step on the way to this longitudinal analysis. We have now included a more thorough description of the overall study methods in the paper and have made it clearer that this analysis will follow, building on the publication of the results of this study (page 6, line 421-425).

Reviewer 2 Report

Thank you for the opportunity of revising this interesting paper.

Although the sample are patients with COPD that have been referred to pulmonary rehabilitation, they have not been under the effect of any intervention yet. In fact, this is a descriptive cross-sectional study of physical activity and sedentary behaviour patterns of 326 patients with COPD recruited from a clinic. Most of these findings have been described previously by Mesquita and co-workers in “Physical activity patterns and clusters in 1001 patients with COPD” Chron Respir Dis. 2017 Aug;14(3):256-269. I understand, some additional analyses compared to the already published paper have been performed and most of the sample is employed (75.3%), probably responsible for some of the differences found, therefore the authors have to be congratulated for their considerable amount of work. But I am uncertain that the take home messages and the implications for pulmonary rehabilitation are novel from those that can be inferred from the paper of Mesquita and co-workers. I have however, revised the manuscript carefully and I leave below my comments/suggestions for your consideration.

Additional aspects for consideration

Introduction

You are presenting a descriptive cross-sectional study without any intervention. At that point in time, your sample is also composed solely by patients with COPD. Therefore, I would recommend removing from the introduction the argument that “So far, there is hardly any evidence as to whether the results of Mesquita et al. can also be applied to the situation of people with COPD before they start PR”. Before they start PR they continue to be patients with COPD.

Methods

Page 2 line 89 “…paid for by”???

In the flow chart (Figure 1) the number and reasons for exclusion are different from the text. Please revise both. In this study there is no T1 however this timepoint is mentioned in page 3 line 99. I also find unnecessary to have a T0 in this cross-sectional descriptive study.

In the characteristics of the sample I think it would be important to add the ABCD and patients’ comorbidities. In fact, I think these two aspects should be included in the clusters analyses. How does the distribution of ABCD and comorbidities occur in each cluster? Is it similar? Is it different? This would be important to further enhance our understanding of your findings.

Page 4 line 119 - I would recommend to report the all interpretation of CAT and not just when it indicates a high Impact.

All patients were contacted 1 month before PR but when were assessments taken? And where? Were patients invited to come to the clinic or were they at home? Was this voluntary or paid? Could this be an additional reason for relatively a low adherence (52.4%) to participate? How many 6MWT were performed and where? I believe these are unclear from your methodology.

Additionally, I was unable to understand which 5 days you used for analyses in the accelerometers. Did you include patients with just week days and patients with week and weekends? How was this taken into account? There is also a need to add a ref for the use of at least 10h wear-time per day – page 4 line 136.

Page 4 line 139 – In the sentence starting with “Across, the entire sample…” please remove the word “study” from “study participants” and “at” from “in at vigorous PA”.

I would recommend you performing the k-medoids for the cluster analysis and reporting the graph produced instead of the scree plot (although the elbow shows the 3 clusters), as I believe it would be more informative to understand how they behave.

Comparisons with ABCD and comorbidities should also be taken into account in your analysis.

Results

Page 5 line 178 – “Against the background of these ambiguous recommendations”… awkward sentence please revise it.

Page 5 line 195 “comparably active” comparably to what? Please be clearer.

Page 6 line 201 – Although I understand what you want to say by “linear relationship” I don’t believe you can actually state this. What about progressive increase from cluster 1 to cluster 4? Just a suggestion but the expression linear relationship has a statistical meaning which does not seem to apply here. For clarity purposes I think it would be of added value to include in table 2 the sitting time and the analysis, this would facilitate the full interpretation of Figure 2.

Page 8 – what happens to the effect of number or type of comorbidities? Or ABCD distribution?

I would recommend always using clusters when referring to the four groups and not groups and clusters.

I don’t think Figure 3 adds anything to what is already presented in Table 2. I recommend adding the other analysis and removing this Figure 3.

Page 8 lines 233-235 sitting time is not represented graphically and I believe it would facilitate the interpretation of the findings. Also please always use the same terminology minutes (place the equivalent in hours).

Discussion

As explained previously  would rephrase your first sentence as they are patients with COPD at that point in time.

Page 10 line 269 – “Against the background of various reviews…” again awkward sentence please revise it.

Page 10 line 278 “show but that---“???

Page 10 line 280 – “… there are subgroups that are remarkably active” this is not a new finding at all. Please reference it appropriately.

Page 10 line 286 – given the complexity of physical activity and SB patters, just based on your findings to state “This suggest that health-related movement might rather be a one-dimensional construct” seems too speculative. Please consider revising it.

Page 10 lines 294-295 – Due to low physical capacity…” – again long bouts of MVPA may be unattainable” What do you mean by long bouts? According to WHO recommendations? Also this can be unattainable due to many reasons and in fact most of your sample has a high exercise capacity… this sentence seems to benefit of a bit more reflexion.

Page 10 line 301 – “h/day” or “h/d”?

Page 11 Cluster differences regarding comorbidities? Number and Type? And ABCD distribution?

Page 11 – Implications for pulmonary rehabilitation. This section has several parts repeated from other sections of the article. I would recommend to remove from one place to avoid repetition of ideas.

Conclusion

I would not “force” the before rehabilitation as this is a cross-sectional study of patients with COPD. Yes, data have been collected before PR but you are not looking at comparisons with before and after but instead describing what is happening in this groups of patients with COPD re PA and SB patterns. I would rephrase slightly the conclusions.

I would recommend always report to patients with COPD or people with COPD and not COPD patients.

Author Response

Reviewer 2:

Thank you for the opportunity of revising this interesting paper.

Although the sample are patients with COPD that have been referred to pulmonary rehabilitation, they have not been under the effect of any intervention yet. In fact, this is a descriptive cross-sectional study of physical activity and sedentary behaviour patterns of 326 patients with COPD recruited from a clinic. Most of these findings have been described previously by Mesquita and co-workers in “Physical activity patterns and clusters in 1001 patients with COPD” Chron Respir Dis. 2017 Aug;14(3):256-269. I understand, some additional analyses compared to the already published paper have been performed and most of the sample is employed (75.3%), probably responsible for some of the differences found, therefore the authors have to be congratulated for their considerable amount of work. But I am uncertain that the take home messages and the implications for pulmonary rehabilitation are novel from those that can be inferred from the paper of Mesquita and co-workers. I have however, revised the manuscript carefully and I leave below my comments/suggestions for your consideration.

—> Dear reviewer, Thank you very much for reading and reviewing our manuscript. We kindly appreciate the evaluation of our article.

In view of your concerns about the novelty of our results, we would like to highlight two points: Firstly, to our knowledge, this is the first study in the field of COPD applying an analytical approach which ensures differentiation between SB (≤ 1.5 MET) and very light forms of PA (1.5 MET < x < 2 MET). This allows a precise analysis of total SB and a distinction from standing. We have now made this clear in the discussion (page 4, line 326-328). Secondly, the overriding goal of the STAR study is a longitudinal group-based analysis of changes in PA and SB after rehabilitation. Therefore, the participants in the STAR study wear the accelerometer again 6 weeks and 6 months after rehabilitation. The presented paper represents the first important step on the way to this longitudinal analysis. We have now included this point in the paper and made it clearer that this analysis will follow. (page 6, line 421-425)

Introduction

You are presenting a descriptive cross-sectional study without any intervention. At that point in time, your sample is also composed solely by patients with COPD. Therefore, I would recommend removing from the introduction the argument that “So far, there is hardly any evidence as to whether the results of Mesquita et al. can also be applied to the situation of people with COPD before they start PR”. Before they start PR they continue to be patients with COPD.

—> We have deleted this sentence as per your recommendation (page 2, line 70).

Methods

Page 2 line 89 “…paid for by”???

—> We have now clarified this sentence: The rehabilitation was paid for by the German pension insurance fund (page 2, line 90).

In the flow chart (Figure 1) the number and reasons for exclusion are different from the text. Please revise both. In this study there is no T1 however this timepoint is mentioned in page 3 line 99. I also find unnecessary to have a T0 in this cross-sectional descriptive study.

—> We have deleted the mention of T0/T1 and have clarified the text and figure differences (page 3, line 99-104).

In the characteristics of the sample I think it would be important to add the ABCD and patients’ comorbidities. In fact, I think these two aspects should be included in the clusters analyses. How does the distribution of ABCD and comorbidities occur in each cluster? Is it similar? Is it different? This would be important to further enhance our understanding of your findings.

—> Thank you for this comment. This information is now also included in table 2 (page 8) 99-104).

Page 4 line 119 - I would recommend to report the all interpretation of CAT and not just when it indicates a high Impact.

—> Thank you, we have included the interpretation of all CAT scores (page 4, line 125).

All patients were contacted 1 month before PR but when were assessments taken? And where? Were patients invited to come to the clinic or were they at home? Was this voluntary or paid? Could this be an additional reason for relatively a low adherence (52.4%) to participate? How many 6MWT were performed and where? I believe these are unclear from your methodology.

—> See line 87 for clarification of how patients were referred to the study/rehabilitation.

See line 117-119 for explanation of when/where PA was measured.

See line 95 for clarification that patients were not paid to be in the study. Also note that the majority of patients were not paying for the rehabilitation as this was covered by their workers pension insurance

See line 126 for clarification around the 6MWT methods.

Additionally, I was unable to understand which 5 days you used for analyses in the accelerometers. Did you include patients with just week days and patients with week and weekends? How was this taken into account? There is also a need to add a ref for the use of at least 10h wear-time per day – page 4 line 136.

—> We have now clarified this section and added in a reference to the COPD specific PA measurement recommendations (page 4, line 147-150).

Page 4 line 139 – In the sentence starting with “Across, the entire sample…” please remove the word “study” from “study participants” and “at” from “in at vigorous PA”.

—> Thank you, we have removed these words (page 4, line 153).

I would recommend you performing the k-medoids for the cluster analysis and reporting the graph produced instead of the scree plot (although the elbow shows the 3 clusters), as I believe it would be more informative to understand how they behave.

—> The suggested k-medoids method belongs to the so-called partitional cluster methods. Here, the researchers are already aware of the number of clusters, trying to optimize the final cluster solution. We, however, wanted to explore the optimal number of clusters. Therefore, we decided for a hierarchical cluster analysis process. Nevertheless, we thank you very much since we noticed that we did not give the full information regarding the selected cluster method. We added this relevant information (page 5, line 161-162).

Comparisons with ABCD and comorbidities should also be taken into account in your analysis.

—> ABCD classifications were taken into account (table 2 and page 8, line 239-242).

Results

Page 5 line 178 – “Against the background of these ambiguous recommendations”… awkward sentence please revise it.

—> This sentence has been re-worded to improve flow and readability (page 5, line 195).

Page 5 line 195 “comparably active” comparably to what? Please be clearer.

—> The sentence has been expanded to include a clearer statement and comparison (page 6, line 213).

Page 6 line 201 – Although I understand what you want to say by “linear relationship” I don’t believe you can actually state this. What about progressive increase from cluster 1 to cluster 4? Just a suggestion but the expression linear relationship has a statistical meaning which does not seem to apply here. For clarity purposes I think it would be of added value to include in table 2 the sitting time and the analysis, this would facilitate the full interpretation of Figure 2.

—> This is a great suggestion. We have made the change to the sentence. Table 2 includes sedentary time. We have now named the line in question more clearly.

Page 8 – what happens to the effect of number or type of comorbidities? Or ABCD distribution?

—> We added this analysis in Table 2 and integrated it into the discussion (page 11, line 358-360).

I would recommend always using clusters when referring to the four groups and not groups and clusters.

—> Thank you, we have edited the manuscript to now include the term clusters throughout.

I don’t think Figure 3 adds anything to what is already presented in Table 2. I recommend adding the other analysis and removing this Figure 3

—> We deleted this figure and added the analysis as recommended.

Page 8 lines 233-235 sitting time is not represented graphically and I believe it would facilitate the interpretation of the findings. Also please always use the same terminology minutes (place the equivalent in hours).

—> Please see Table 2 and Figure 2 where we have outlined average sedentary time per day.

Discussion

As explained previously would rephrase your first sentence as they are patients with COPD at that point in time.

—> This sentence has been refined.

Page 10 line 269 – “Against the background of various reviews…” again awkward sentence please revise it.

—> This sentence has been refined (page 9, line 290).

Page 10 line 278 “show but that---“???

—> This sentence has been reformulated (page 10, line 300).

Page 10 line 280 – “… there are subgroups that are remarkably active” this is not a new finding at all. Please reference it appropriately.

—> This sentence has been refined, and references previous similar findings (page 10, line 299-301).

Page 10 line 286 – given the complexity of physical activity and SB patters, just based on your findings to state “This suggest that health-related movement might rather be a one-dimensional construct” seems too speculative. Please consider revising it.

—> This sentence has now been deleted. As you mention this is speculative and does not add to the sentiment being described above (page 10, line 307-308).

Page 10 lines 294-295 – Due to low physical capacity…” – again long bouts of MVPA may be unattainable” What do you mean by long bouts? According to WHO recommendations? Also this can be unattainable due to many reasons and in fact most of your sample has a high exercise capacity… this sentence seems to benefit of a bit more reflexion.

—> This sentence has been reformulated (page 10, line 316-320).

Page 10 line 301 – “h/day” or “h/d”?

—> Changes have been made here to align the findings.

Page 11 Cluster differences regarding comorbidities? Number and Type? And ABCD distribution?

—> We discussed the cluster differences for the number of clusters and the ABCD distribution (page 11, line 358-360; see also your comment regarding table 2 on page 9).

Page 11 – Implications for pulmonary rehabilitation. This section has several parts repeated from other sections of the article. I would recommend to remove from one place to avoid repetition of ideas.

—> This section has been refined. We moved some of the repeated parts to other sections and erased the doublings (page 11, line 387; page 12, line 401; page 12, line 404).

Conclusion

I would not “force” the before rehabilitation as this is a cross-sectional study of patients with COPD. Yes, data have been collected before PR but you are not looking at comparisons with before and after but instead describing what is happening in this groups of patients with COPD re PA and SB patterns. I would rephrase slightly the conclusions.

—> Thanks for your suggestion, we have rephrased this statement (page 13, line 449).

I would recommend always report to patients with COPD or people with COPD and not COPD patients.

—> This has been revised throughout.

Round 2

Reviewer 2 Report

I believe the manuscript has been significantly improved from the previous version. Not totally convinced of the novelty of the findings with this cross-sectional study and what it adds to the already known in the body of literature but the authors are to be commended by their hard work.